# Genetic Variation in Antioxidant Response Modulates the Level of Oxidative Stress in Youth with Type 1 Diabetes and Poor Glycemic Control

**DOI:** 10.3390/antiox11091726

**Published:** 2022-08-31

**Authors:** Anita Morandi, Massimiliano Corradi, Chiara Zusi, Claudia Piona, Silvia Costantini, Marco Marigliano, Claudio Maffeis

**Affiliations:** Section of Pediatric Diabetes and Metabolism, Department of Surgery, Dentistry, Pediatrics, and Gynecology, University of Verona, Piazzale Stefani, 37126 Verona, Italy

**Keywords:** *NFE2L2*, rs2364723, oxidative stress, type 1 diabetes

## Abstract

Background: The minor allele of the single nucleotide polymorphism (SNP) rs2364723 of *NFE2L2*, a gene encoding a master antioxidant transcription factor, has been associated with poor cardiovascular outcomes and with complications of type 2 diabetes. We assessed the association between rs2364723 of *NFE2L2* and oxidative stress in children/adolescents with type 1 diabetes (T1D). Methods: In 384 children/adolescents with T1D (age 15.7 ± 3.2 years, 207 males), we assessed the oxidative stress by measuring the concentration of derivatives of reactive oxygen metabolites (d-ROMs) and we genotyped the rs2364723 SNP by real time polymerase chain reaction. Results: The concentration of d-ROMs was 372.8 ± 64.6 Carratelli units. The minor genotype (CC) of rs2364723 at *NFE2L2* was associated with higher concentration of derivatives of d-ROMs in the subgroup with HbA1c ≥ 8% (B = 47.85, p for genotype ∗ HbA1c interaction = 0.019). Conclusions: The carriers of the minor genotype of rs2364723 may have increased oxidative stress compared to their counterparts with other genotypes, especially in case of poor glycemic control. This observation needs to be replicated and confirmed in larger independent cohorts of youth with T1D.

## 1. Introduction

Type 1 diabetes (T1D) is characterized by micro- and macrovascular morbidity and mortality [1,2,3,4,5]. Oxidative stress, a well-known driver of inflammation/vascular damage and an accelerator of T1D complications in animal models, is increased in youth with T1D. Current and past glycemic control account for a limited portion of the interindividual variability of the oxidative stress of patients [6,7]. Improving our knowledge of what determines oxidative stress in T1D is of major importance since it would contribute to reveal the bases of the so-called “residual risk”, i.e., the vascular risk not accounted for by metabolic control and traditional risk factors. Therefore, novel knowledge about oxidative stress in T1D would provide novel potential therapeutic targets beyond glycemic control.

Genetic variation in the antioxidant response and its interaction with the glycemic control may explain part of the inter-individual variability of the oxidative stress of patients with T1D. Nuclear factor erythroid 2-related factor 2 (NRF2), encoded by *NFE2L2*, triggers a master first line response against oxidative stress because it enhances the transcription of tens of genes involved in the oxides neutralization, in response to a plethora of environmental or endogenous deviations in redox metabolism [8]. The suppression or activation of the NFR2 pathway has proved to favor or to inhibit the development of complications in animal models of T1D [6].

The genetic variation at *NFE2L2* has been associated with vascular function (forearm blood flow), autonomic function (heart rate variability), major vascular events and type 2 diabetes complications in cohorts and case control sets of adults [9,10,11,12,13]. Especially, the minor allele of rs2364723, which is in complete linkage disequilibrium with the promoter SNP rs35652124, known to impede the NFE2L2 transcriptional activity, has been consistently associated with unfavorable outcomes [9,10,11,12,14,15]. We have previously observed that the concentration of serum derivatives of reactive oxygen metabolites (d-ROMs) is increased compared to healthy peers in a cohort of 412 children/adolescents with T1D [7]. Of these 412 patients, 384 were genotyped at the rs2364723 locus, and the minor genotype showed increased d-ROMs without statistical significance [7]. We have subsequently hypothesized that the genetic variation at *NFE2L2* may influence the level of oxidative stress mainly when oxidative stressors are increased and the NFE2L2-triggered antioxidant response is required, like in the case of poor glycemic control. Therefore, we reanalyzed the variation in oxidative stress in relation to the *NFE2L2* rs2364723 genotype, assessing if genotype and glycemic control interact in predicting the level of oxidative stress in children/adolescents with T1D.

To the best of our knowledge, no study has tested the interaction between *NFE2L2* locus variation and glucose control as predictor of oxidative stress in people with T1D.

The aim of this brief communication is to report the results of the post hoc analysis we performed to assess if rs2364723 at the *NFE2L2* locus and glycemic control interact in predicting oxidative stress in our cohort of children/adolescents with T1D.

## 2. Materials and Methods

We studied 384 children/adolescents with T1D (age 3.6–23.5 years) followed up at the Regional Center for Pediatric Diabetes of the University Hospital of Verona, Italy. The inclusion/exclusion criteria and the detailed study protocol have been previously reported [7]. All participants have had diabetes for at least two years and were out of partial remission, defined as insulin dose-adjusted glycated hemoglobin A1c (HbA1c) (HbA1c% + 4 × insulin dose (U/kg/day) ≤ 9%) [16].

HbA1c was measured with Cobas b101 (Roche, Switzerland) by immunoturbidimetric assay. Derivatives of reactive oxygen metabolites (d-ROMs) concentration was measured with a commercial kit (Diacron, Italy). The derivative species were measured with a spectrophotometer giving a broad absorbance peak at ~505 nm. The color intensity obtained is directly proportional to the ROMs concentration in the sample. The measuring unit is the Carratelli unit (U-Carr) (1 U-Carr = 0.08 mg H2O2/dL). The genotype of rs2364723 was determined using pre-designed TaqMan probes (Applied Biosystem, Waltham, MA, USA), according to the manufacturers’ protocol, with QuantStudio TM 5 Real Time polymerase chain reaction (RT-PCR) (Applied Biosystem, Waltham, MA, USA. Assay identity: C___351878_10).

The day of d-ROMs and HbA1c measurements, all patients had a physical examination with the measurement of height and weight to determine the z-score of the body mass index (z-BMI) according to the World Health Organization (WHO) charts, and to exclude any acute illness [17].

All the parents/guardians of the participating children and adolescents and the participants above 18 years of age signed an informed consent to participate in the study which was approved by the Ethical Committee of the University Hospital of Verona.

According to their HbA1c, patients were divided into two groups: one under the value of 8% and the other with a value ≥8%, with 8% representing the median of the whole sample.

By general linear model, we determined if the rs2364723 genotype and the HbA1c category interact in predicting the concentration of d-ROMs, independently of gender, age and z-BMI. All the analyses were performed by IBM SPSS Statistics 24 package (IBM statistics).

## 3. Results

The concentration of d-ROMs was 372.8 ± 64.6 U-Carr. The rs2364723 genotypes were distributed as follows: GG: 210, G/C:147, CC: 27 in the total sample, GG: 116, G/C:87, CC: 15 in patients with HbA1c < 8% and GG: 94, G/C:60f, CC: 12 in patients with HbA1c ≥ 8%). The genotypes were in Hardy–Weinberg equilibrium in the whole sample and in the HbA1c categories separately and the genotype distribution did not differ according to the genotype category (*p* = 0.76). Z-BMI (B = 11.29, *p* = 0.001) and being boys under 12 years of age (B for age ∗ gender interaction = 41.32, *p* = 0.018) predicted d-ROMs concentration, as previously reported [7]. The HbA1c category and rs2364723 were not associated with d-ROMs but their interaction was associated (Table 1 and Figure 1). In fact, patients who carried the rs2364723 CC genotype had a significantly higher concentration of d-ROMs compared to GG/GC carriers, in the subgroup with HbA1c ≥ 8% (B = 47.85, *p* = 0.019) (Table 1 and Figure 1).

## 4. Discussion and Conclusions

In our cohort of youth with T1D, the minor genotype of rs2364723 at the *NFE2L2* locus increased the overall systemic oxidative stress by 47.85 U-carr, corresponding to 0.74 standard deviation among patients with poor glycemic control (HbA1c ≥ 8%). Despite this large effect size, it may be argued that the described interaction is not very relevant because the subgroup of patients with minor genotype and poor glycemic control included only 12 patients, corresponding to the 3.1% of the whole cohort. However, if confirmed in larger cohorts, the evidence of a deleterious effect of the minor genotype of rs2364723 on the oxidative stress of patients with poor glycemic control would have a significant relevance, especially in clinical contexts with a high prevalence of patients with poor glycemic control. The CC genotype of rs2364723 has been previously associated with major cardiovascular events and reduced heart rate variability [9,13]. Similar to our results, the minor allele of rs2364723 has been associated with the risk of complications in patients with type 2 diabetes of both Chinese and European ancestry [11,12]. To the best of our knowledge, this is the first study assessing the association between rs2364723 and any outcome in patients with T1D. It is plausible that a redox challenging condition like poor glycemic control amplifies the genetic inter-individual variability of any response induced by oxidative stress, like NRF2 cascade. The observed interaction is a proof of concept that genetic variability may significantly modulate the redox balance in patients with T1D, especially when glycemic control is poor. This implies that genetics, probably besides lifestyle variables, contributes to explain the slice of inter-patient variability in the oxidative stress that is not explained by glycemic control. Several NRF2 inducing drugs have been developed in recent years and have been tested with promising results in phase II and phase III trials against many chronic diseases accompanied by oxidative stress and against the complications of type 2 diabetes [8]. One of these molecules, dimethyl fumarate, has been approved by the Food and drug Administration for the treatment of multiple sclerosis and psoriasis [8]. As it has been established that NRF2 delays the progression of T1D complications in animal models, T1D is currently regarded as an ideal target for NFR2 enhancing therapies and it is expected that the number of clinical trials employing NFR2 inducing molecules in patients with T1D will rapidly multiply [6]. Once replicated in larger independent cohorts, the result of the present study would highlight that the patients carrying the minor genotype of rs2364723 represent a small subgroup that might mostly benefit from NFR2-inducing drugs, especially during periods of poor glycemic control.

## Figures and Tables

**Figure 1 antioxidants-11-01726-f001:**
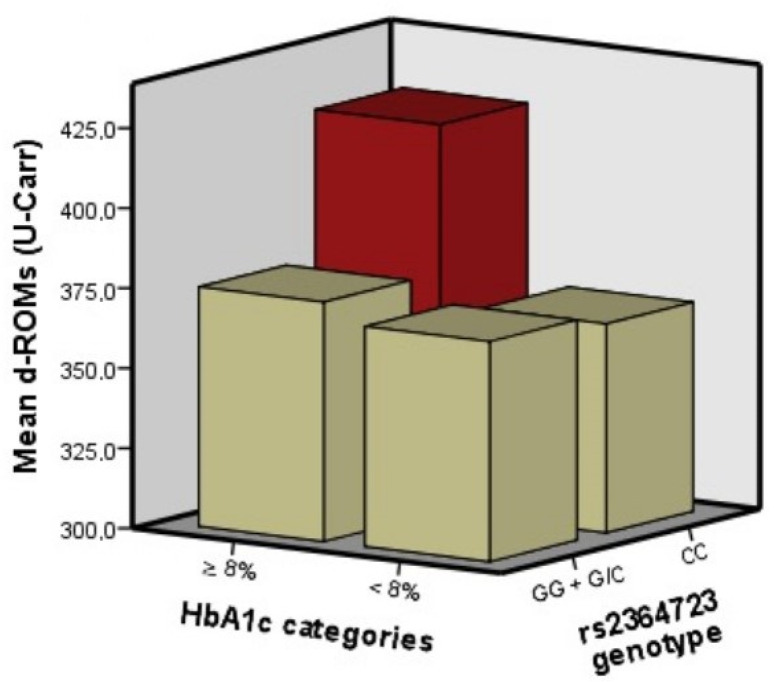
Concentration of d-ROMs according to the category of glycemic control and to the rs2364723 genotype.

**Table 1 antioxidants-11-01726-t001:** Characteristics of the study participants according to their HbA1c category, their genotype at rs2364723.

	HbA1c < 8%			HbA1c ≥ 8%			P for HbA1c Category	P for Genotype	P for HbA1c Categories + Genotype
	Genotype at rs2364723			Genotype at rs2364723					
	GG + GC (n = 203)	CC (n = 15)	Total (n = 218)	GG+GC (n = 154)	CC (n = 12)	Total (n = 166)			
**d-ROMs (U-carr)**	368.3 (63.5)	364.5 (43.6)	368.1 (62.1)	374.7 (65.7)	421.1 (64.9)	377.7 (66.5)	0.12	0.15	0.019
**Age (years)**	15.6 (3.7)	15.8 (3.1)	15.6 (3.7)	15.8 (3.1)	14.9 (4.0)	15.7 (3.2)	0.78	0.11	0.12
**M/F**	115/88	10/5	125/93	76/78	6/6	82/84	0.15	0.35	0.72
**z-BMI**	0.38 (1.0)	0.23 (0.7)	0.37 (0.9)	0.38 (0.9)	0.49 (0.9)	0.39 (0.9)	0.61	0.91	0.40
**Disease duration (years)**	8.2 (4.2)	8.1 (3.6)	8.2 (4.1)	8.4 (3.8)	8.0 (3.6)	8.4 (3.8)	0.27	0.76	0.89

HbA1c = Glycated A1c Hemoglobin; d-ROMs = derivatives of reactive oxygen metabolites; U-carr = Carratelli unit; M/F = males/females; z-BMI = z-score of body mass index.

## Data Availability

The datasets generated during and/or analyzed during the current study are not publicly available but are available from the corresponding author on reasonable request.

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
