# Peer review of "Genetic Variation in Antioxidant Response Modulates the Level of Oxidative Stress in Youth with Type 1 Diabetes and Poor Glycemic Control"

_antioxidants, 2022, doi:10.3390/antiox11091726_

Round 1

Reviewer 1 Report

I have reviewed the manuscript antioxidants-1858065 entitled ”Genetic variation in antioxidant response modulates the level of oxidative stress in youth with type 1 diabetes and poor glycemic control”.

The authors analyzed the interaction between NFE2L2 rs2364723 polymophism, and glycemic control in predicting oxidative stress in children/adolescents with type I diabetes. The article may provide useful information about the genetic influences in diabetes type 1. I suggest major revision.

1.     The entire article need typing revision.

2.     Introduction lines 23-27 – it is a very complex phrase. I suggest splitting in 2 phrases.

3.     The aim of the study should be shorten and very clear.

4.     The present communication is a part of a larger study? The authors should mention this or they should include inclusion and exclusion criteria

5.     The results presented in lines 77-80 should be detailed. Also the genotypes should be also analyzed in patients with a good glycemic  control. There is a difference between the groups regarding genotypes?

6.     Instead of conclusions, I suggest to entitled this part of the article discussion and conclusion. The conclusion should be very short. The first part of this section represents actually discussion

Author Response

Reviewer 1:

I have reviewed the manuscript antioxidants-1858065 entitled ”Genetic variation in antioxidant response modulates the level of oxidative stress in youth with type 1 diabetes and poor glycemic control”.

The authors analyzed the interaction between NFE2L2 rs2364723 polymophism, and glycemic control in predicting oxidative stress in children/adolescents with type I diabetes. The article may provide useful information about the genetic influences in diabetes type 1. I suggest major revision.

We thank the Reviewer for the overall positive comment.

  1. The entire article need typing revision.

The article was edited by an English mother tongue copy editor.

  1. Introduction lines 23-27 – it is a very complex phrase. I suggest splitting in 2 phrases.

We followed the Reviewer’s suggestion.

  1. The aim of the study should be shorten and very clear.

We shortened and better clarified the aim of the study.

  1. The present communication is a part of a larger study? The authors should mention this or they should include inclusion and exclusion criteria

We clarified that the present communication is part of a larger study (lines 52-56 and 75-76).

  1. The results presented in lines 77-80 should be detailed. Also the genotypes should be also analyzed in patients with a good glycemic There is a difference between the groups regarding genotypes?

We addressed the Reviewer’s suggestion in lines 104-108 and in the novel Table 1. All the patients were genotyped, and the categories of glycaemic control did not differ as regards the genotype distribution.

  1. Instead of conclusions, I suggest to entitled this part of the article discussion and conclusion. The conclusion should be very short. The first part of this section represents actually discussion.

We followed the Reviewer’s suggestion, renaming the paragraph “Discussion and conclusion”.

Reviewer 2 Report

The authors genotyped 384 patients with type 1 diabetes (T1D) for a common single nucleotide polymorphism (SNP) at the NFE2L2 locus (rs2364723) and investigated the association of this SNP with serum concentration of derivatives of reactive oxygen metabolites (d-ROMs). They found that the minor allele of this SNP was associated with higher levels of d-ROMs in T1D patients with worse glycemic control (with glycated haemoglobin > 8%).

These are interesting results that suggest that patients with lower antioxidant activity, due to the genetic variant at NFE2L2, are more susceptible to the effects of oxidative stress caused by poor glycemic control.

The manuscript is concise and well written. Although it is presented as a “brief report”, the manuscript would benefit from additional data.

1) Abstract. A short and concise abstract is always appreciated, but in this case, the authors may have exaggerated. A more structured abstract that includes the implications of the findings would be beneficial to the readers.

2) It is a convention that gene symbols should be in italic. Please use italics for NFE2L2 when referring to the genetic locus or gene, and non-italic when referring to the NFE2L2 protein.

3) Define abbreviations (e.g. HbA1c, WHO, BMI, FDA, etc) unless the journal stipulates otherwise.

4) The minor allele has been previously associated with unfavourable outcomes. As the SNP is located outside of the NFE2L2 coding region, it would be useful to say something (in the introduction or discussion) about how this SNP leads to lower antioxidant activity.

5) Figure 1 is very useful as it gives an immediate visual impression of how the d-ROMs vary between the subgroups. However, it does not allow a more detailed analysis of the data. A table would be an additional benefit, with more complete data for each analysed group along with intervals of dispersion.

6) The results summarised in the conclusion should be consistent with those in the results section. The statement in the conclusions “increased the overall systemic oxidative stress by 0.74 standard deviation…”, may confuse readers because this is a different reporting format than that presented in the results section.

7) Consider replacing the section heading “Conclusions” for “Discussion” and check the numbering of the headings (section 3 Results is followed immediately by section 5 Conclusions, rather than #4).

8) Material & Methods. Please complete manufacturer details for Applied Biosystem and provide the id of the genotype assay. Please inform the statistical software that was used.

Author Response

Reviewer 2:

The authors genotyped 384 patients with type 1 diabetes (T1D) for a common single nucleotide polymorphism (SNP) at the NFE2L2 locus (rs2364723) and investigated the association of this SNP with serum concentration of derivatives of reactive oxygen metabolites (d-ROMs). They found that the minor allele of this SNP was associated with higher levels of d-ROMs in T1D patients with worse glycemic control (with glycated haemoglobin > 8%).

These are interesting results that suggest that patients with lower antioxidant activity, due to the genetic variant at NFE2L2, are more susceptible to the effects of oxidative stress caused by poor glycemic control.

The manuscript is concise and well written. Although it is presented as a “brief report”, the manuscript would benefit from additional data.

We thank the Reviewer for the overall positive comment.

  • A short and concise abstract is always appreciated, but in this case, the authors may have exaggerated. A more structured abstract that includes the implications of the findings would be beneficial to the readers.

The new version of the manuscript includes a structured abstracts providing the implications of the finding.

  • It is a convention that gene symbols should be in italic. Please use italics for NFE2L2 when referring to the genetic locus or gene, and non-italic when referring to the NFE2L2 protein.

We apologize for the inaccuracy. We made the proper corrections.

  • Define abbreviations (e.g. HbA1c, WHO, BMI, FDA, etc) unless the journal stipulates otherwise.

We defined all the abbreviations in the abstract, main text and novel Table 1.

  • The minor allele has been previously associated with unfavourable outcomes. As the SNP is located outside of the NFE2L2 coding region, it would be useful to say something (in the introduction or discussion) about how this SNP leads to lower antioxidant activity.

The studied SNP is in complete linkage disequilibrium with a functional SNP impeding the NFE2L2 transcriptional activity. We explained this at lines 50-52 (novel references 14 and 15).

  • Figure 1 is very useful as it gives an immediate visual impression of how the d-ROMs vary between the subgroups. However, it does not allow a more detailed analysis of the data. A table would be an additional benefit, with more complete data for each analysed group along with intervals of dispersion.

We inserted Table 1 to enable a more detailed analysis of the data.

  • The results summarised in the conclusion should be consistent with those in the results section. The statement in the conclusions “increased the overall systemic oxidative stress by 0.74 standard deviation…”, may confuse readers because this is a different reporting format than that presented in the results section.

We better clarified the effect size of the described interaction at lines 119-121 of the discussion.

  • Consider replacing the section heading “Conclusions” for “Discussion” and check the numbering of the headings (section 3 Results is followed immediately by section 5 Conclusions, rather than #4).

We made the appropriate changes according to the Reviewer’s suggestions.

  • Material & Methods. Please complete manufacturer details for Applied Biosystem and provide the id of the genotype assay. Please inform the statistical software that was used.

We added the information requested by the Reviewer.

Reviewer 3 Report

Paper by Morandi et al. analyze the interaction among rs2364723 minor allele , Hb1Ac levels and derivatives of reactive oxygen metabolites on a cohort of 384  type 1 diabetes patients concluding that minor allele of rs2364723 at NFE2L2, was associated with higher concentration ofd-ROMs in the subgroup with HbA1c ≥ 14. results are presented extremely aggregate  and a reader cannot analyze the relevance of data presented. 

Considering the consistent number of patients studied, tables detailing group and subgroup results and differences appear necessary. 

In addition I would suggest that comparison with a group of age and sex matched controls should be made.

Finally the conclusion section should be  considered a discussion section and a discussion on the role of rs2364723 SNP presented in this paper on T1D respect T2D published papers might be useful. 

Author Response

Paper by Morandi et al. analyze the interaction among rs2364723 minor allele , Hb1Ac levels and derivatives of reactive oxygen metabolites on a cohort of 384  type 1 diabetes patients concluding that minor allele of rs2364723 at NFE2L2, was associated with higher concentration ofd-ROMs in the subgroup with HbA1c ≥ 14. results are presented extremely aggregate  and a reader cannot analyze the relevance of data presented.

Considering the consistent number of patients studied, tables detailing group and subgroup results and differences appear necessary.

We thank the Reviewer for the suggestion. We provided disaggregated data analysis in the new Table 1 and we commented on the potential relevance of the reported result at lines 121-128 of the discussion.

In addition I would suggest that comparison with a group of age and sex matched controls should be made.

We compared the oxidative stress of our cohort with T1D with that of a cohort of healthy peers, in a previous larger study the present short communication is part of (see reference 7).  We demonstrated significantly higher concentration of d-ROMs in the cohort with T1D. Of course, the control cohort would not be suitable to assess the role of the rs2364723 according to the HbA1c level.

Finally the conclusion section should be  considered a discussion section and a discussion on the role of rs2364723 SNP presented in this paper on T1D respect T2D published papers might be useful.

We renamed the conclusion paragraph “Discussion and conclusion” and we discussed the role of rs2364723 in T2D at lines 130-132, also adding a very recent reference about this SNP and CKD in T2D (new reference 12).

Round 2

Reviewer 1 Report

I agree with the new version of the manuscript with a single observation. The authors should write a clear conclusion at the end of the article. 

With this recommendation the article could be published in Antioxidants journal.

Reviewer 2 Report

The authors have significantly improved the manuscript. Only one minor detail: in table 1, the numbers within parenthesis are not defined. They probably represent standard deviations, but there is no sure indication of this. Perhaps a short explanation in the footnote would be enough.